# Coupling of the AQUATOX and EFDC Models for Ecological Impact Assessment of Chemical Spill Scenarios in the Jeonju River, Korea

**DOI:** 10.3390/biology9100340

**Published:** 2020-10-19

**Authors:** Jaehoon Yeom, Injeong Kim, Minjeong Kim, Kyunghwa Cho, Sang Don Kim

**Affiliations:** 1School of Earth Sciences and Environmental Engineering, Gwangju Institute of Science and Technology, Gwangju 61005, Korea; duawogns12@gist.ac.kr; 2Jeonbuk Department of Inhalation Research, Korea Institute of Toxicology, Jeongeup 56212, Korea; injeong.kim@kitox.re.kr; 3School of Urban and Environmental Engineering, Ulsan National Institute of Science and Technology, Ulsan 689-798, Korea; paekhap0835@naver.com (M.K.); firstkh@gmail.com (K.C.); 4Center for Chemicals Risk Assessment, Gwangju Institute of Science and Technology, Gwangju 61005, Korea

**Keywords:** chemical accident, toluene, ecological assessment, biomass, AQUATOX, EFDC

## Abstract

**Simple Summary:**

This study proposed a methodology to simulate ecological damage in a toluene spill situation by coupling AQUATOX, an established ecological assessment model, and EFDC, a Lagrangian fluid diffusion model. TheAQUATOX-EFDC simulation showed a significant ecological impact, especially the greatest damage on the fish species group, the top predators.

**Abstract:**

In this study, an ecological impact was assessed for the short-term leak scenario through the AQUATOX-EFDC model, which combines the proven ecological model AQUATOX with the hydrodynamic model EFDC. A case study of the coupled AQUATOX-EFDC model was conducted for 30–30,000 kg toluene leak scenarios in the Jeonju River in South Korea. A 21-day scenario simulation was conducted, and the impact of the toluene spill accident was evaluated by comparing the biomass between the control simulation and the perturbed simulation. As a result of the simulation, it was found that in the scenario in which 3000 kg of toluene was leaked for a day, a substantial change was expected in the range of 0–640 m from the accident site. Additionally, for a 30,000 kg leak, a substantial change was expected in the range of 0–2300 m from the accident site, and the greatest damage was observed for the fish species group, the top predators. As a result, the AQUATOX-EFDC simulation showed a significant ecological impact, and the proposed model will be helpful to understand the ecological impact and establish the management strategy for the ecological risk of the chemical spill.

## 1. Introduction

Increasing production and use of chemicals has resulted in increased numbers of the chemical accidents around the world [1,2]. Large chemical spills from chemical plants such as the Bhopal disaster have taught people that they need chemical management to protect from chemical accidents [3]. Three main chemical accidents categories, which are explosions, fires, and leaks, have been identified [1,2,4]. Among these, leakage accidents can have both short-term and long-term impacts on ecosystems due to the spread of chemicals in the environment [5]. Therefore, the impact management of leak accidents is one of the important challenges for environmental management.

South Korea, a manufacturing-based industrial country, also produces a large amount of chemicals every year, and the number of chemical accidents is also increasing every year. The Korea National Institute of Chemical Safety (NICS) has recorded that hazardous chemical spill accidents increased from 14 accidents per year to 72 accidents per year from 2008 to 2017 in South Korea [6] Since chemical accidents possibly cause non-reversible damage to the environmental system, the government of South Korea also intended to introduce an assessment system for the impact of accidents [7].

In order to prevent and manage chemical accidents, NICS designated 97 chemicals as accident preparedness substances to control usage and disposal. One of the benzene, toluene, ethylbenzene, and xylenes (BTEX) chemical group, toluene, was also classified as the most hazardous pollutant and included in the accident preparedness substances list. Toluene has very low solubility, acute toxicity, and genotoxicity [8]. Additionally, many studies have reported the toxicity of toluene on aquatic organisms such as algae [9,10], fish, and invertebrates [11,12]. The quantity of toluene transfer is 1.696 × 10^4^ tons per year in South Korea in 2016 [6], as toluene has been broadly used in the industrial sector of South Korea. Therefore, there is a possibility that toluene leakage chemical accidents can occur at any time. In this study, a toluene spill scenario will be simulated to assess the impact of chemical spills on ecology.

Since there are various exposure routes to the receptors in the real environment, traditional risk assessment methodology using laboratory toxicity test data on single species is not enough to be considered ecological realism [13]. As a substitute, the integrated ecological model can simulate the complex interaction of the real environment. AQUATOX has been used for predicting various environmental contamination scenarios for lakes, rivers, and reservoirs [14,15,16,17]. This model not only considers general content of water quality such as ammonia, phosphate, dissolved oxygen, total organic carbon, chlorophyll a, and temperature, but the interaction between organisms in the food web including diatoms, invertebrates, and fishes of aquatic systems.

AQUATOX has been applied to the toxicity assessment of various chemical substances in the water system. In the Tajan River, diazinon was assessed by using annual monitoring data of the river [14]. The model was also used for calibration and validation of nitrobenzene monitoring data in the Songhua River [15]. In addition, the ecological impact of polycyclic aromatic hydrocarbons was assessed by biomass change of organisms in the lake [16]. AQUATOX was also used for evaluating ecosystem service delivery with insecticide impacts on a freshwater lake [18]. However, it was not applied to the hypothetical spill scenarios in the river yet because the performance of the model is not enough to simulate dynamic conditions in chemical spills. Non-steady state dynamics in the ecosystem and hydrodynamics for the chemical spill should be considered together because leakage accidents usually occur within a short time.

In this study, AQUATOX was linked to the external model, EFDC (environmental fluid dynamic source code) which is a hydrodynamic model developed by US EPA (The United States Environmental Protection Agency) [19], to augment the limits in the hydrodynamic simulation. The aims of this study were (1) developing the methodology of impact assessment for the chemical spill situation in the river and (2) applying the developed impact model to the toluene accidental spill scenario in the Jeonju River, South Korea.

## 2. Materials and Methods

### 2.1. Concept of AQUATOX-EDFC Model

AQUATOX is an ecological risk assessment model for evaluating the fate of common contaminants. This model not only considers general contents (ammonia, phosphate, dissolved oxygen, chlorophyll a, temperature, and so on) of water quality but the interaction between organisms in the food web including algae, invertebrates, and fishes in aquatic systems [18]. Since AQUATOX uses the food web model, it can simulate a causal relationship between all the contents in the water system mechanically. AQUATOX simulates all constituents of the water body as the concentration is in contrast with the population model [20]. A model needs a lot of input data, which cover the dynamics of the water body, nutrient composition of the system, and ecological food web. However, the AQUATOX model is inadequate to simulate hydrodynamic flow in the stream. Additional model simulation is necessary to acquire realistic hydrodynamic flow information. In contrast, EFDC is a hydrodynamic model, which uses differential transport equations. The EFDC model is able to estimate the inflow rate and water level of the segment. Therefore, AQUATOX-EFDC model can use dynamic flow and water level data from the EFDC model as an input variable of each segment of the AQUATOX model independently. The AQUATOX-EFDC model was operated by inserting the output of segment water flow into the following segment with the continuous flow function (Figure 1). This simulation was for estimation of biomass change of each species and toluene concentration in each segment of the chemical accident site at time t. Each segment of the model was progressed by the independent box model with boundary conditions.

### 2.2. Characteristics of Study Area

Jeonju is a city in South Korea which has a population of 656,117 and an industrial area whose pollutant transfer is 3.35 × 10^6^ kg per year in 2017 [6]. The Jeonju River is a branch of the Geum River, South Korea. The Jeonju River passes through Jeonju industrial area where a dye factory is located. Due to the adjacent industrial area of the river, hazardous chemical transporting vehicles often pass through the river with the risk of chemical spills.

The location of the Jeonju River and segments were used for an ecological assessment in the chemical accidental scenario (Figure 2). The environmental ecological data was collected from the Water Environment Information System (WEIS, http://water.nier.go.kr/) operated by the Korea Ministry of Environment (Appendix A), and they measured all of data at S1 point on 1 July 2014. This study focused on the stretch starting at S1 in the Jeonju River to 2.86 km downstream from the starting point (S6). The surface area and length of each segment was measured using Google Maps (Appendix A).

### 2.3. Species Matching and Food Web

In the AQUATOX model, the system requires more than 20 parameters that reflect the ecological and physiological characteristics of each species. Interactions in ecosystems are calculated based on the input parameters of each species. The ecosystem of the Jeonju River is composed of 4 fish, 30 invertebrates, and 14 diatoms, which are mostly comprised of Korean native species (Appendix A). Since the information on domestic species was not enough to complement the required parameter for the AQUATOX simulation, the species in the Jeonju River were approximated as those present in the AQUATOX library. In order to match species with the library, taxonomy, ecology, and morphology of species were considered. Taxonomical similarity means morphological and ecological similarity. In the AQUATOX model, the ecological niche of species was important to calculate interaction within the food web and the mean weight of species can be used for estimating the ecological niche. Thus, species matching considered (1) taxonomical fitness, (2) feeding habits, and (3) mean weight (*W*) of species [21]. The average weight of each species (*W*) was derived from Equation (1) [22].
(1)$$W=aL^{b}$$
where *a* and *b* are inherent coefficients for each species, and *L* is the body length of the species (Appendix A) [23,24,25,26,27,28,29,30]. The species with 3 criteria matching with the AQUATOX library resulted in 3 fishes, 8 invertebrates, and 6 diatoms (Appendix A). During the AQUATOX simulation, there is an additional limitation on the number of species that can be simulated at once [20]. So that, to reduce the number of species, the second matching was processed. The species with low population densities were matched with other species whose population densities were larger when there was a high similarity between different species. The biomass of combined species was defined as the sum of the biomass of each species. Finally, 13 species (3 fish, 7 invertebrates, and 3 diatoms) were selected to simulate the chemical spill in the Jeonju River, and the food web was constructed (Appendix A). Additionally, the 3 diatom species were divided into 2 compartments, periphyton and phytoplankton. Each diatom species is simulated with 2 compartment linkages.

### 2.4. Biomass Density of Species

The obtained biomass of species was converted into dry mass density which is a suitable form for AQUATOX. Dry biomass density of diatom (*d_dry.diatom_*, g·m^−2^) is calculated by following Equation (2).
(2)$$d_{dry.diatom}=\frac{d_{cell}\times f_{diatom}}{0.36}$$
*d_cell_* is the cell number density of diatom (cell/m^2^), and *f_diatom_* is 0.01625 ng C/cell as a conversion factor of carbon content of a diatom cell (ng C/cell), which is for Bacillariophyceae studied in Korean water systems [31]. Additionally, 0.36 is the conversion factor for carbon contents to dry weight of diatom cells [32].

In the case of invertebrates and fish, the wet biomass density was converted to the dry mass following Equations (3) and (4). Wet biomass density of species *i* (*d_wet,i_*, g/m^2^) was calculated by *Wt_avg_* and number density (*d_N_*, m^−2^) by Equation (3).
(3)$$d_{wet,i}=d_{N,i}\times Wt_{avg,i}$$
where *d_N_* is the number density (m^−2^) of species *i*. Dry biomass density of species *i* ($d_{dry,i}$, g/m^2^) was calculated using the following equation:(4)$$d_{dry,i}=d_{wet,i}\times f_{\frac{d}{w},i}$$
where *f_d/w,i_* is the dry weight/wet weight ratio of species *i* (g/g, Appendix A).

### 2.5. EFDC Modeling for Flow Rate of Each Segment

In the AQUATOX model, each segment has two hydrodynamic input values; input volume flow rate and output volume flow rate. EFDC modeling was applied to generate the flow dynamics of each segment. In this study, EFDC explorer version 8.2 (DSI, LLC, Washington, DC, USA) was used for constructing the initial hydrodynamic state of the Jeonju River.

In the simulation of EFDC, the Jeonju River was built based on the river planning document suggested by the Jeonju local government. The number of the curvilinear-orthogonal grids was 1457, and the average orthogonal deviation was 1.27°. The average length of grids was 47 m of dx and 115 m of dy. The j axis consisted of 4 grids for the Jeonju River and 14 grids for the Mankyeong River (Appendix A). The *x* axis of the grid means the horizontal cross-sectional direction of the river, *y* axis of the grid means the flow direction of the river, and j axis means the depth of the river. The boundary condition of the EFDC simulation was set as shown in Appendix A. The daily flow rate was collected from the Water Environmental Information System (WEIS). The flow rate series was sorted by value in descending order. The 95th, 185th, and 275th flow rates (m^3^/s) were defined as annual high, median, and low water flow states of the site. Bottom roughness and wall roughness were used for calibrating the model. The hydrodynamic simulation was conducted for the annual median water flow of the Jeonju River. Finally, the result from EFDC simulation was converted into the inflow rate of the segment which was defined as the x direction velocity of the central grid in the left boundary of the segment.

### 2.6. AQUATOX-EFDC Coupling

The assignment of EFDC output into AQUATOX was complemented by the simultaneous equation. In this study, the input flow rate volume in AQUATOX was calculated based on the time series output of the water level obtained from the EFDC. The water levels of the segment division in the AQUATOX simulation of central grids at the start and end boundary of each segment resulted from the EFDC simulation. The water level series was converted into water volume by
(5)$$V_{i,t}={\bar{WL}}_{i,t}\times A_{i}$$
where *V_i,t_* is the volume of the segment *i* at time *t*, ${\bar{WL}}_{i,t}$ is the average water level of segment *i* at time *t*, and *A_i_* is the surface area of segment *i. A_i_* is measured by the ranging function of Google Maps. Volume change of segment *i* at time *t* ($a_{i,t}$) is the same as the difference between inflow rate and outflow rate. Thus, $a_{i,t}$ can be calculated as follows [33]:(6)$$a_{i,t}=\frac{dV_{i,t}}{dt}=V_{i,t+1}-V_{i,t}=b_{i,t}-b_{i+1,t}$$
where $b_{i,t}$ is the volume inflow rate into segment *i* at time *t*. $b_{i,t}$ can be defined as Equation (7).
(7)$$b_{i,t}=u_{i,t}\times W_{i}\times D_{i}$$
where $u_{i,t}$ is the inflow rate into segment *i* at time *t*, *W_i_* is the width of segment *i*, and *D_i_* is the water depth of segment *i*. The simultaneous equations for each segment were transferred into matrix form as shown in Equation (8) with the boundary condition of volume in starting segment $b_{1}$. $b_{i,t}$ was calculated as input inflow rate volume for the AQUATOX model simulation.
(8)$$\left(\begin{array}{ccc} b_{1,1} & \ldots & b_{i,1}\\ \vdots & \ddots & \vdots \\ b_{1,t} & \ldots & b_{i,t} \end{array}\right)=\left(\begin{array}{ccc} a_{1,1} & \cdots & a_{i-1,1}\\ \vdots & \ddots & \vdots \\ a_{1,t} & \cdots & a_{i-1,t} \end{array}\begin{array}{c} b_{1,1}\\ \vdots \\ b_{1,t} \end{array}\right){\left(\begin{array}{ccc} \begin{array}{cc} 1 & 0\\ -1 & 1 \end{array} & \begin{array}{cc} \ldots & \ldots \\ 0 & 0 \end{array} & \begin{array}{cc} 0 & 1\\ 0 & 0 \end{array}\\ \begin{array}{cc} \vdots & -1\\ \vdots & 0 \end{array} & \begin{array}{cc} \ddots & \ddots \\ \ddots & \ddots \end{array} & \begin{array}{cc} 0 & \vdots \\ 0 & \vdots \end{array}\\ \begin{array}{cc} 0 & 0\\ 0 & 0 \end{array} & \begin{array}{cc} 0 & 0\\ \ldots & \ldots \end{array} & \begin{array}{cc} 1 & 0\\ -1 & 0 \end{array} \end{array}\right)}^{-1}$$

Since this study assessed the short-term impact of chemical spills for the local river, simulations assumed environmental factors from the land around the river (rainfall, non-point pollution, etc.) do not have a substantial impact during the simulation period. Thus, simulation only considered isolated river environments except for hydrological water distribution from the land. In addition, it was supposed that the entire section of the river has the same initial environmental conditions due to the relatively short length of the region.

The AQUATOX model simulated linked the multi-segment project. In the multi-segment simulation, spatial variation between segments was considered by dividing segment as distance. The mass balance between parameters including trophic interaction, hydrodynamic transfer, photosynthesis, degradation, and chemical reaction was maintained during the multi-segment simulation [33]. The model of the Jeonju River was constructed by 5 segments. In the all segments, initial water quality, the food web, and biomass of species were applied identically for all segments. During the AQUATOX-EFDC simulation, each segment operated the mass balance equation for all variables simultaneously. The calculated concentration of toluene and water quality of each segment were transferred into input data of the next segment by the linkage function. For the smooth simulation of the hypothetical Jeonju River in the AQUATOX model, it was assumed that (1) there is a segment having a large water volume, (2) its initial environmental condition including biomass density is the same as the initial environmental condition of the target stretch, and (3) this supplied the ongoing upstream inflow into the target segment with controlled condition.

The flow of medium and materials between segments were entered by the mass balance equation of each segment with a non-steady state equation with a 1 h time step and all the linkages between segments were set as “cascade” mode which is the mode to describe one-sided flow between segments. [34]. Time series of the flow rate data from EFDC modeling were entered by “cascade” option into “show link data” tap [34].

### 2.7. Simulation of Control and Perturbed Scenario

In South Korea, the Toxic Chemical Control Act set up the maximum carrying capacity of toxic chemicals. From this regulation, the transportation which carries over the 3000 kg of toluene at one time should be reported and monitored by the local environmental agency. The amount of toluene leaked in the accident scenario was determined based on the Toxic Chemical Control Act (Appendix A). It was assumed that the toluene leak begins at Segment B and the leak is set to end 24 h after the start of the spill. The amount of toluene released per scenario for the perturbed simulation was set using “Dissolved org. tox 1” option in the AQUATOX model. The toluene concentration and the water volume input were made using the “Dynamic loading” function in the “Inflow loading” option [34]. Additionally, toxicity data of each species is collected from ECOTOX and ICE (Interspecies Correlation Estimation) website (Appendix A) [12,35]. Simulations were operated in the control simulation mode and the perturbed simulation mode for each scenario. During the simulation, the toxicant in the perturbed scenario affected biomass of organisms and detritus. The toxic effect of toluene on the ecosystem converted the biomass of living organisms to biomass of detritus during simulation [33].

### 2.8. Calculation of Relative Biomass and Impact Indicators

Simulation results provided time series of biomass for each species in the Jeonju River model. Biomass data were interpreted by converting into relative biomass change. Equation (9) was used for calculating relative biomass change. Relative biomass change was defined as the ratio between the change of biomass after the perturbed scenario and the biomass result from the control simulation.
(9)$$\varepsilon {\;}_{day}\left(\%\right)=\frac{\sum_{t=0}^{day}B_{iPERT}-\sum_{t=0}^{day}B_{iCONT}}{\sum_{t=0}^{day}B_{iCONT}}\times 100$$

Additionally, relative variation of average biomass ($\bar{\omega }$) was calculated as the relative variation between the average biomass of accidental simulation (${\bar{B}}_{iACC}$, g_dry_/m^2^) and the average biomass of control simulation (${\bar{B}}_{iCONT}$ g_dry_/m^2^) for given species *i* [36].
(10)$${\bar{\omega }}_{i}\left(\%\right)=\frac{{\bar{B}}_{iACC}-{\bar{B}}_{iCONT}}{{\bar{B}}_{iCONT}}\times 100$$

Overall average biomass relative variation is the mean of $\bar{\omega }$ for *N* living compartments of the ecosystem [36]:(11)$$\bar{\omega }\left(\%\right)=\overset{N}{\underset{i=1}{\sum }}\frac{{\bar{B}}_{iACC}-{\bar{B}}_{iCONT}}{{\bar{B}}_{iCONT}}\times 100\times \frac{1}{N}$$

In this study, the ecosystem was judged to be perturbed when the biomass indicator changed by 5% or more relative to the control simulation.

## 3. Results and Discussion

### 3.1. Toluene Concentration after Spill Accident in Jeonju River

Toluene concentration in the Jeonju River was simulated by the AQUATOX-EFDC model. Figure 3 shows the toluene concentration in the water system is increased by 10 times for every 10-fold increase of the quantity in the toluene spill. The maximum toluene concentration in the river is simply proportional to the quantity of spilled concentration.

The maximum toluene concentration was drastically decreased by the increase of distance from the accident segment. Toluene concentration in the water body was maximized in one day after the spill and occurred to be eliminated in two days after spilling with a half-life time of 1.9 h. Toluene concentration for 5–20 days after the toluene spill is about zero. The highest toluene concentration in each segment from 30 to 30,000 kg of the toluene spill are 1.46 × 10^2^–1.46 × 10^5^ µg/L in segment B, 2.06 × 10–2.06 × 10^4^ µg/L in segment C, 4.81–4.81 × 10^3^ µg/L in segment D, and 2.27–2.27 × 10^3^ µg/L in segment E for each scenario.

Among existing studies, there have been many attempts to simulate changes in the concentration of organic substances in the environment through modeling of oil leakage, one of the common pathways in which toluene is leaked into water [37]. Various existing models of oil spills often assumed the transport of oil through the formation of slicks and the dissolution through oil droplets [37,38,39]. However, the AQUATOX model does not consider slick or droplet models for poorly soluble organic substances such as aromatic oils when calculating the transport of toxic substances. The AQUATOX model does not carry out detailed modeling on the transport of poorly soluble organic substances, so there was a limit to the modeling of environmental leakage of aromatic oil-derived substances such as toluene [33]. However, the results obtained from the calculation of the simple diffusion and transport model of toluene through the coupling of EFDC and AQUATOX showed some agreement with the results of the conventional oil transport model. Among the existing oil spill simulation models, the specific study of oil spill in the river with consideration about slick conformation showed the half-life of toluene was about 2 h [39]. Even though there was no assumption about slick modeling in the AQUATOX, the elimination trend of toluene was similar to a more practical model. The simulated half-life of spilled toluene was consistent with the previous study, which gives reliable results in terms of exposure time to actual receptor organisms.

### 3.2. Total Biomass Changes of Organisms in the Jeonju River

The model simulated a toluene spill with initial conditions of species biomass and water quality of the Jeonju River at 1 July 2014. Perturbed scenarios were designed for 30 kg, 300 kg, 3000 kg, and 30,000 kg of toluene spill scenarios. As a result of the simulation of the toluene leakage accident, the impact of the spill on the total biomass reduction in the Jeonju River was found to be weak for most scenarios, and only for scenario in which 30,000 kg of toluene was spilled was a strong change caused (Figure 4).

Comparing the figures for each scenario, for scenarios 1 and 2, there was no strong effect in all areas of the Jeonju River ($\bar{\omega }<5\%,\;\varepsilon {\;}_{20\;days}<5\%)$. On the other hand, in scenario 3, the change in the total biomass was weak ($\varepsilon {\;}_{20\;days}=-$1.29% < 5%), but the overall average biomass relative variation ($\bar{\omega }$) was strongly changed which is calculated as −21.06% for Segment B, that indicates there is a large change in the composition of the species. Additionally, there were no strong changes in both the total biomass and the composition of species for segments C, D, and E. For the harshest condition, a leak of 30,000 kg, in Segment B, $\bar{\omega }$ was −31.5% and $\varepsilon {\;}_{20\;days}$ was calculated as −7.16%, showing a substantial change in the total biomass. It also had a great influence on the composition. In this scenario, in the downstream Segment C, $\bar{\omega }$ is −23.5%, $\varepsilon {\;}_{20\;days}$ is −1.29%; in Segment D, $\bar{\omega }$ is −9.04%, $\varepsilon {\;}_{20\;days}$ is −0.68%; in Segment E, $\bar{\omega }$ is −5.11%, $\varepsilon {\;}_{20\;days}$ is −0.61%. It shows strong effects on the composition of species even in the segments C, D, E of the Jeonju River, but the total biomass does not appear to change substantially.

Therefore, based on the total biomass indicator, only 30,000 kg of toluene leakage scenario showed substantial damage to the ecosystem, and the affected area was also limited to 560 m from the accidental spill site, which is close to the accidental spill site. However, based on the composition of the species indicator, the ecosystem was strongly affected from the 3000 kg toluene leakage scenario. In scenario 3, it affected up to 640 m from the accidental spill site, and in scenario 4, it affected up to 2.30 km from the accidental spill site.

For the toluene spill in the Jeonju River, the risk is predicted for situations that correspond to scenario 4 based on the total biomass, and the risk is expected for situations that are based on scenario 3 and 4 based on the average relative biomass change. According to the actual number of leaks during transport by tank lorries in Korea, the number of leaks of 30 kg or less for the last seven years (2014–2020) for all chemical substances was 60. The number of leaks of 30–300 kg or L was 23, the number of leaks of 300–3000 kg or L was 14, the number of leaks of 3000–30,000 kg or L was 11, and finally the number of leaks of more than 30,000 kg or L was counted as 0 (2020, NICS). Therefore, there is a possibility that spills larger than scenario 3 can occur, which is expected to cause substantial damage to the ecosystem when toluene is actually transported around the Jeonju industrial complex. Additionally, this study hypothesized that there was no pipeline spill and chronic contamination from ground water in the Jeonju river because it was focused on short-term accidental spills.

### 3.3. Biomass Changes of Single Species

The toluene leaks had different effects on individual species. As a result of the simulation of AQUATOX-EFDC, the biomass changes after the accidental spill scenario of each species did not simply decrease, but both increased and decreased depending on the species (Figure 5).

Based on the results of biomass change in Segment B, which is the closest to the point of the accidental spill and where the largest change is shown, the change of absolute biomass density over time for each single species in the Jeonju River shows different trends for each species and differently contributes to the overall average relative biomass variation ($\bar{\omega }$) (Figure 5). The fish groups (minnow, golden shiner, and stoneroller) have the most sensitive response to toluene spills. The minnow species is substantially affected by toluene spillage scenarios of over 300 kg and its relative variation (${\bar{\omega }}_{minnow}$) reaches −86.1% in the harshest scenario. Golden shiner and stoneroller species are also strongly affected by toluene spillage scenarios of over 3000 kg and their variation increases by up to −83.7% (${\bar{\omega }}_{golden\;shiner}),$ and −79.4% $({\bar{\omega }}_{stoneroller}$) in scenario 4 (Figure 5).

The invertebrate groups (mayfly, caddisfly, Odonata, Asian mud snail, riffle beetle, isopod, and chironomid) tend to be affected less by the spilling scenarios (Figure 5). Most sensitive species in the invertebrate group were isopod species which are strongly impacted by toluene spills above 300 kg. Furthermore, the average relative biomass variation (${\bar{\omega }}_{isopod}$) of isopods was increased up to −82.4% in the most severe scenario (Figure 5). Additionally, riffle beetle and chironomid species which are the second most sensitive group in the invertebrate groups were substantially damaged in only the 30,000 kg toluene spilling scenario. Relative variation values of riffle beetle reached up to −21.0% (${\bar{\omega }}_{riffle\;beetle}$) and chironomid reached up to −59.8% (${\bar{\omega }}_{chironomid}$) (Figure 5). On the other hand, the other group (mayfly, caddisfly, Odonata, and Asian mud snail) showed only small changes ($\left|\bar{\omega }\right|<5\%$) in even the harshest simulation condition. Therefore, only three species (isopod, riffle beetle, and chironomid) of the invertebrates group were affected by toluene spills even in the most severe scenario condition.

Among the invertebrate species, especially, mayfly, Odonata, Asian mud snail showed slightly increasing tendency ($\bar{\omega }>0)$ of biomass density after the spill corresponding to the control. For the exceptional trend of mayfly, the result can be analyzed with a simple predator-prey relationship [39]. The increment of mayfly was caused by decreasing numbers of minnow which are predators of mayfly and preservation of *Navicula*, the main food for the mayfly (Figure 5, Appendix A). Asian mud snail, on the other hand, does not have predators in the food chain. The largest prey of this species is detritus, which is calculated from the biomass of the dead creatures in the AQUATOX model (Appendix A). Deaths of species induced by higher than 30,000 kg toluene leakage caused the production of detritus, which resulted in an increased number of Asian mud snails (Figure 5).

The three diatoms species (*Melosira*, *Navicula*, and *Nitschia*) belonging to the diatoms group suffered only small impacts in all the accidental scenarios. *Melosira* species shows $\bar{\omega }$ values from 0.05% to 0.74% in scenarios 1 to 4. Additionally, *Navicula* species showed $\bar{\omega }$ values from 0.05% to 0.43% in scenarios 1 to 4. Additionally, *Nitschia* species showed $\bar{\omega }$ values from 0.03% to 0.98% in scenarios 1 to 4 (Figure 5). Therefore, all the diatom species are perturbed weakly.

Consequently, the average biomass relative variation ($\bar{\omega }$) analysis suggested fish group species are expected to be the most impacted species from a toluene spill scenario in the Jeonju River. Additionally, diatom species and most of invertebrate species were safe even in severe spilling. In detail, for scenario 1, there were no affected species. Additionally, for scenario 2, minnow and isopod were two impacted species. Additionally, for scenario 3, minnow, golden shiner, stoneroller, and isopod were four affected species. Finally, in the most severe condition, scenario 4, minnow, golden shiner, stoneroller, isopod, riffle beetle, and chironomid were six strongly affected species.

However, it should be highlighted that the condition of the scenarios was only for momentary leakage for 1 d and not continuous leakage. In other words, even brief exposure to toluene caused by leakage can bring ecosystem damage.

### 3.4. Relationship between Biomass Changes and LC50 of Organisms

The simulated biomass changes of species in the Jeonju River showed a different tendency compared to a laboratory-scale toxicity test, which is defined as median lethal concentration (LC50) value (Figure 6). Although fishes and diatoms have similar LC50, the biomass decrease of fishes was more severe than that of the diatom. Ecological, or trophic, effects on the higher predator fish species are greater than the toxic effect of toluene upon producer diatoms. When toluene is leaked into the Jeonju River ecosystem, even if the higher predators have the same LC50 value, the survival was remarkably low due to lack of food and deficient nutrients. The biomass of the upper predators was affected by the reduction of both the lower stage predators and producers, so it is more sensitive than the toxic values would suggest. In addition, diatoms at the bottom of the food chain had slightly higher biomass in the accident simulation than the control simulation. This was because the physiology of each species in the food chain is different, and the impact of the spill is an opportunity for faster-growing lower trophic levels to increase their biomass [40] when predation pressure is reduced.

Even species with a similar level of sensitivity to toxicity can be affected differently by the same scenario. This study suggests that the vulnerability of the aquatic organism is not only related to sensitivity to toxicant but also to complementary effects of changes within the food web. Thus, the AQUATOX-EFDC model can provide more realistic estimates of biomass changes considering interactions within ecosystems than the classical toxicological index (e.g., LC50) for the chemical spill situations.

### 3.5. Impact of Toluene Spill on Ecological Structure

The biomass difference between the control and the accident simulations was calculated as the average biomass change relative to control ($\varepsilon {\;}_{day},\;\%$) in each species group. From the point of view of total average relative biomass ($\varepsilon {\;}_{day},\;\%$), only scenario 4 substantially affected total biomass of the Jeonju River ($\varepsilon {\;}_{day}$ > 5%) (Figure 4). On the other hand, the overall average biomass relative variation ($\bar{\omega }$) showed scenario 3 and 4 made substantial biomass perturbation (Figure 4). The difference between two quotients was from different biomass proportion of species group (Figure 7). The fish species group had a very small proportion (<5%) of the total biomass but fish were more sensitive to toluene spills compared to other species groups. Invertebrate and diatom species represented more than 90% of the total biomass in the ecosystem whereas their biomass change after the spill was relatively small ($\varepsilon {\;}_{day}$ < 5%) except in scenario 4 (Appendix A and Figure 7).

The structural change of the ecosystem in the Jeonju River was obvious in the fish species group. In scenarios 3 and 4, the biomass proportion of the fish group to total biomass was decreased by up to 68% and 85% for each scenario corresponding to the control scenario (Figure 7). On the other hand, invertebrate and diatoms species showed increasing biomass proportion to total biomass after the spill. Especially, the biomass proportion of invertebrate species continuously increased in scenarios 3 and 4. It is caused by rapid elimination of fish species which are the predators of invertebrates. Even in the most severe scenario 4, the biomass of invertebrates initially decreased briefly under the influence of toxic substances and then continued to increase, exceeding the increase in scenario 3 after 20 d. The increasing of biomass proportion of invertebrates was mainly induced by a decrement of predator group (fish group).

Therefore, in the accidental toluene spilling scenario, the toxic effect of toluene release is maintained only until the toluene is banished, which is quite rapidly, but pressure of structural change from ecosystem perturbation is expected to persist (Figure 3 and Figure 7).

## 4. Conclusions

This study proposed a methodology to simulate ecological damage in a chemical spill situation by coupling AQUATOX, an established ecological assessment model, and EFDC, a Lagrangian fluid diffusion model. In this study, we simulated the time of the spill as July 1. We assumed the water flow, biomass, and water quality (e.g., nutrient concentrations) to be stable, using the values observed on 1 July 2014. The actual water system is different because the state of the water body is constantly changing, but this is a short-term simulation over the duration of which we assume the physical water system to change little. As a result of the simulation, it was found that in the scenarios in which 30 kg and 300 kg of toluene was leaked for a day, there would be no strong damage to the area of the Jeonju River. For a 3000 kg leak, a strong change was expected in the range of 0–640 m from the accident site, and for a 30,000 kg leak, a strong change was also expected in the range of 0–2300 m from the accident site, and the greatest damage was observed for the fish species group, the top predators. Accidental spills of toluene and chemicals may affect the ecosystem damage during long and short periods. In this study, there is no longer period simulation for predicting the recovery of the ecosystem from the toluene spill. Future study with longer-term simulation would be expected to discover recovery of the ecosystem after chemical spills. Due to the lack of sufficient ecological data which is necessary for the AQUATOX model, it is difficult to accurately predict qualitative biomass changes, but it is expected to be used for prediction of vulnerable species and accident ranges in accident situations. The methodology of this study can be used not only for toluene, but also for industrial materials that can be the results of chemical accidents, so it can be used for predicting the impact after a spill in situations where it is possible to estimate the amount of spillage in the event of a chemical accident.

## Figures and Tables

**Figure 1 biology-09-00340-f001:**
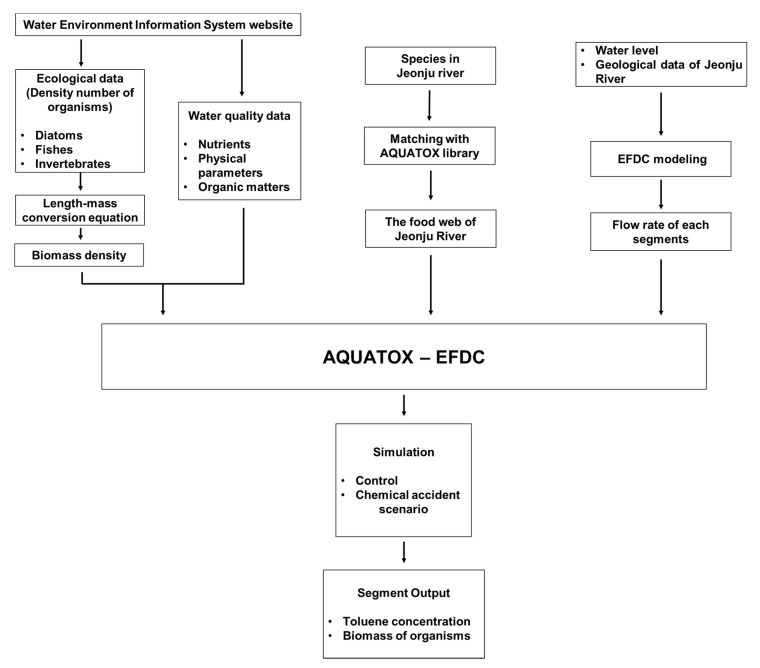
Framework of AQUATOX-EFDC model for an ecosystem assessment in chemical spill scenario.

**Figure 2 biology-09-00340-f002:**
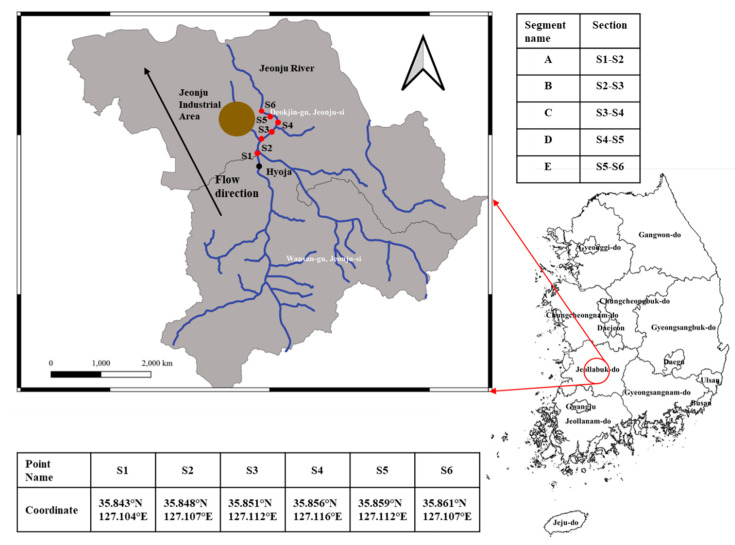
Location of Jeonju River.

**Figure 3 biology-09-00340-f003:**
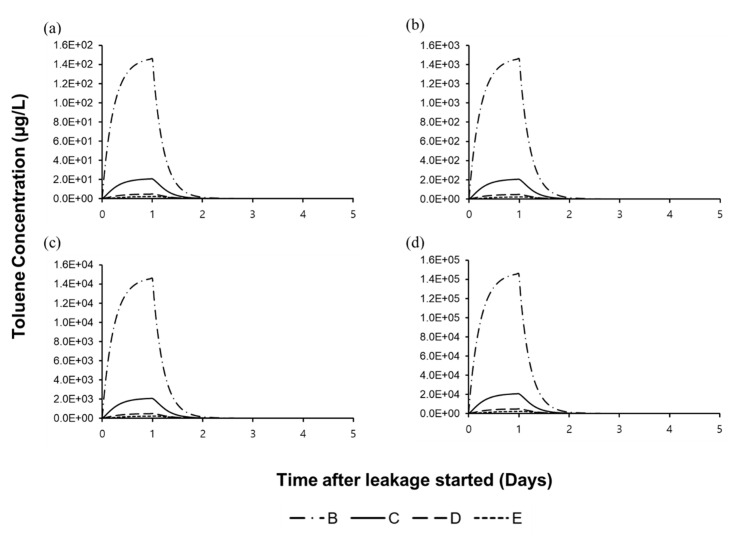
Concentration of toluene in the Jeonju River for (**a**) 30 kg spill, (**b**) 300 kg spill, (**c**) 3000 kg spill, and (**d**) 30,000 kg spill of toluene. Lines B, C, D, E means the name of each segment.

**Figure 4 biology-09-00340-f004:**
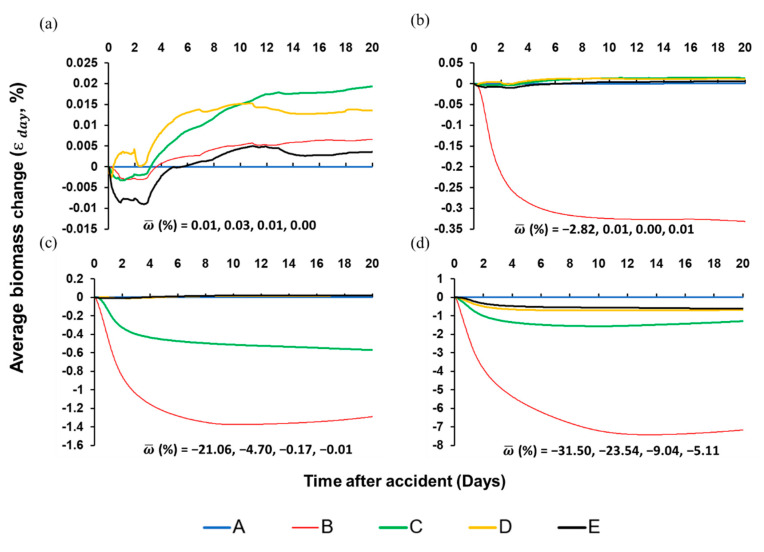
Average biomass change ($\varepsilon {\;}_{days}$, %) of total biomass of organisms in the Jeonju River for (**a**) 30 kg spill, (**b**) 300 kg spill, (**c**) 3000 kg spill, and (**d**) 30,000 kg spill of toluene. Each different color line designates each segment A, B, C, D, and E. Average biomass change (%) means relative change of total biomass of organisms in each segment.

**Figure 5 biology-09-00340-f005:**
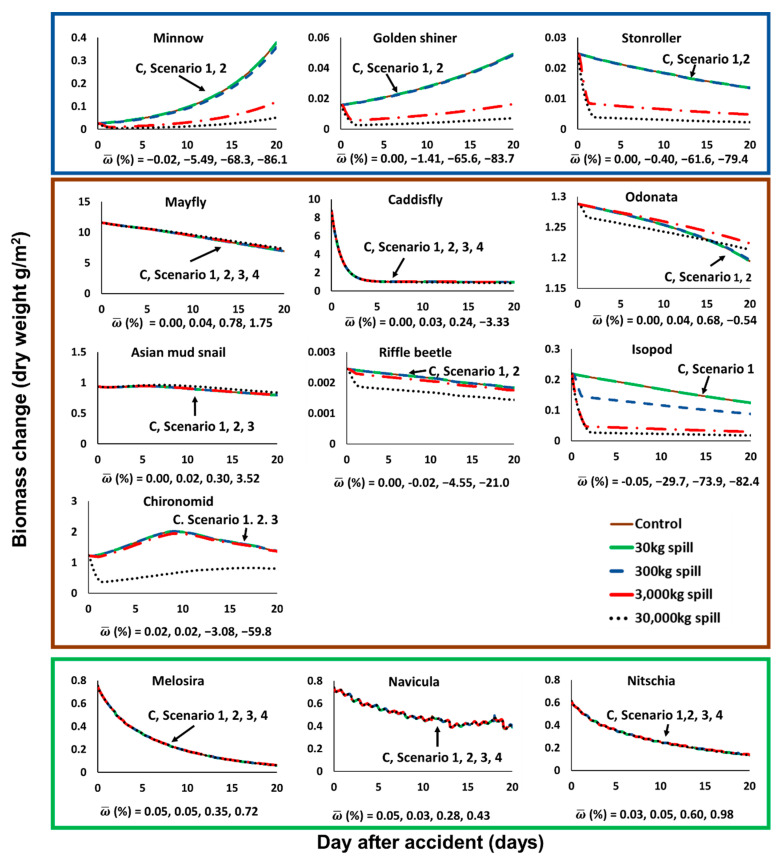
Biomass change (g/m^2^) of each species at segment B of the Jeonju River by toluene spill accident (blue box: fish, brown box: invertebrates, green box: diatoms). Each different color line designates each control, 30 kg, 300 kg, 3000 kg, and 30,000 kg spill scenario.

**Figure 6 biology-09-00340-f006:**
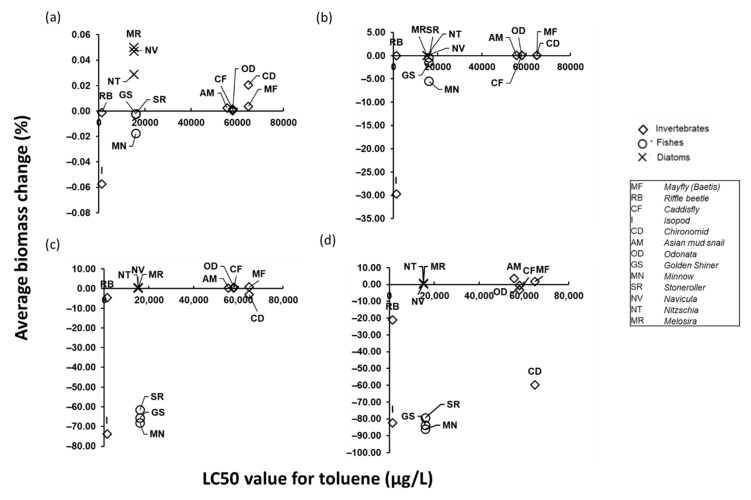
Average biomass change (%) in the Jeonju River versus LC50 (µg/L) value for each species for (**a**) 30 kg spill, (**b**) 300 kg spill, (**c**) 3000 kg spill, and (**d**) 30,000 kg spill of toluene in segment B.

**Figure 7 biology-09-00340-f007:**
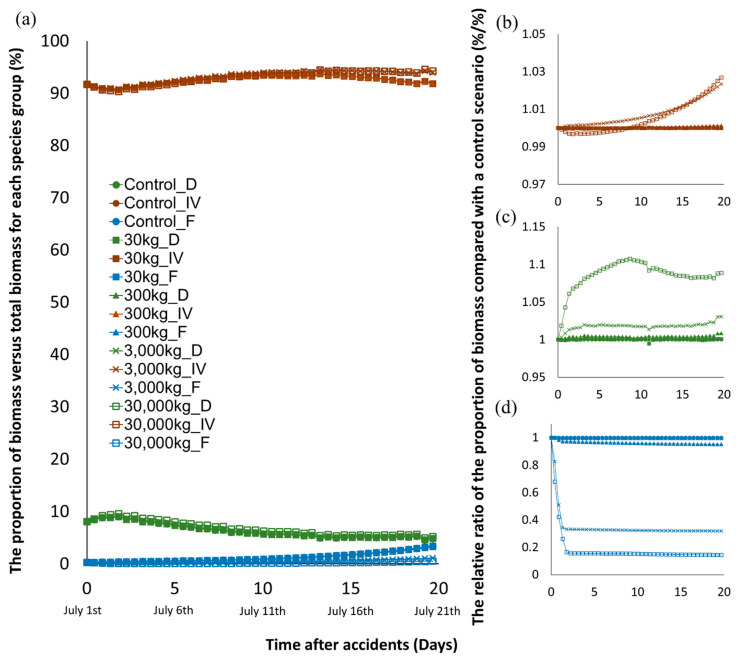
(**a**) The proportion of biomass of each species group versus total biomass (%) (D for diatoms, IV for invertebrates, and F for fishes) for scenarios at segment B. Additionally, the relative ratio of the proportion of biomass in the perturbed scenario versus the proportion in the control scenario (%/%) of (**b**) invertebrates group, (**c**) diatoms group, and (**d**) fishes group.

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
