# Peer review of "Coupling of the AQUATOX and EFDC Models for Ecological Impact Assessment of Chemical Spill Scenarios in the Jeonju River, Korea"

_biology, 2020, doi:10.3390/biology9100340_

Round 1
Reviewer 1 Report
This manuscript reports an ecological impact assessment for the short-term leak accident scenario through the AQUATOX-EFDC model, which combines the proven ecological model AQUATOX with the hydrodynamic model EFDC. Generally, the set of experiments is well planned, the collection of data is appropriate, the analysis is valid, the results are clearly presented, and the discussion is well elaborated and supported by references. I thick this manuscript deserves publication after addressing some language and editing problems.
Specific comments are:
FALL IN SCOPE WITH JOURNAL
Subject fall well within the scope of the journal
NEW AND ORIGINAL
This is an original study since it combines the proven ecological model AQUATOX with a hydrodynamic model EFDC to provide estimates of ecological impact in the Jeonju River.
TITLE AND CONTENTS
Ok, it reflexes what is presented in the manuscript.
ABSTRACT
Ok.
KEYWORDS.
Ok.
INTRODUCTION
OK.
OBJECTIVES
The objective of the study is clearly stated.
MATERIALS AND METHODS
The set of experiments is well planned, the methods and analysis are clearly explained. Some minor comments in this section are:
Line 41: Add "the" before "Korean Institute…"
Lines 41-43: Badly formulated sentence, check the language!
Line 43. "chemical accident" should be plural.
Line 46: "chemical accident" should be plural.
Line 46-47: This is a badly formulated sentence.
Line 47: Should be "One of benzene, toluene, ethylbenzene, and xylenes (BTEX) group chemicals,…
Line 49: What do the authors mean by " and included in accident substance"
Line 50: should be "have reported"
Line 63: should be "AQUATOX has been applied to .."
Line 67: Which research group in USA? This sentence should be rewritten.
Line 73: Environmental fluid dynamic source code (MFDC) should be define here.
Lines 89-90: Is there any reference about EFDC model?
Line 102-103: What is the reference to that release per year?.
Line 128: a, b and L should be in italics.
Line 144: …calculated "by" the following equation.
Line 145: dcell fdiatom should be in italics
Line 150: (dwet,i, g/m2) and Wtavg should be in italics
Line 156: fd/w,i should be in italics
Line 183: Vi,t should be in italics
Line 241: Please, explain why was is set 5%?
RESULTS and DISCUSSION
The results are clearly presented. The discussion is Ok.
TABLES AND FIGURES
Tables and figures are clear.
REFERENCES
Ok
ENGLISH
There are some language problems that should be addressed through out the manuscript.
Reviewer 2 Report
Review of “Coupling of AQUATOX and EFDC model for ecological impact assessment of chemical spill accident scenarios in the Korean river (Jeonju River)” by Yeom et al.
Synopsis:
The authors apply a coupled hydrodynamic model (EFDC) and ecological, food web model (AQUATOX) to simulate the impact of a chemical spill (toluene) within a temperate riverine ecosystem. A range of chemical spill sizes over a 1-day period were simulated. The food web model works by applying a lethal concentration factors to each living group in the model food web and removing biomass. More detail on how the toxic effects were modeled is needed (see below). Impacts were evaluated in terms of the change in living group biomasses relative to a control with no chemical spill. Simulations were conducted for adjoining river segments for which independent and identical descriptions of the food web were modeled.
This is an interesting paper and should be useful a useful guide for other modeling efforts for estimating the short-term effects of toxic spill events in temperate river systems. There are a few issues that should be addressed before publication:
--- Most importantly, how does AQUATOX simulate the physiological response of living groups to toluene? As LC50 values for different living groups are supplied in the Supplemental Material, the reader logically assumes that AQUATOX uses some lethal concentration function that transfers biomass from living biomass-pools to the detritus. This needs to be stated explicitly in the Methods text. What sort of curve is applied for the concentration effects -- linear, exponential, logarithmic? Does AQUATOX apply any other toxicity effects to growth rate, growth efficiency, diet preferences, etc.?
--- The paper should sate explicitly where the reader can go to find the AQUATOX software. Is it available to the public?
--- The English grammar of the text needs proofing and correcting throughout, but the meaning of the text is generally understandable. I’ve made a few specific suggestions, but the full paper needs thorough proofing for grammar.
--- The term “significance” is used to describe the relative scale of the toluene effect on the ecosystem and not to any statistical test. This causes some confusion, and the authors should use a qualitative term such as “strong” or “substantial” in place of “significant”.
--- In the conclusions section, the authors should address (briefly) whether or not toluene (or other chemical classes) may have longer term toxic effects that this short simulation does not consider. The authors could also suggest a set of future simulations that use their model system to examine food web recovery times following chemical spills.
SPECIFIC COMMENTS
TITLE
- Line 2: Consider rearranging word order, “Coupling of the AQUATOX and EFDC models for ecological impact assessment of chemical spill scenarios in the Jeonju River, Korea”
ABSTRACT
- Line 16 and throughout the paper: Consider using “spill” and “accidental spill” instead of “accident”. For example, “In this study, ecological impact was assessed for short-term accidental spills using AQUATOX-EFDC model scenarios, …”
- Line 26: Awkward wording “the distortion of ecosystems remained”. Are the authors referring to time for recovery or to the scale of the impact?
INTRODUCTION
- Line 51: Do you mean 1.696×10^4 tons per year?
METHODS
- Line 80: Please name the “considers general contents of water quality”.
- Line 166: Please define the x, y, and j axes here. The x-axis is defined later at Line 174.
- Line 182: Equation 5 needs to be moved up a couple of lines to this position.
- Line 200: This is the first use of the term “mass-balance”. Please describe what this term refers to. It is the mass-balance of the food web?
- Line 213: Please describe what is meant by “cascade mode” for readers who do not use the AQUATOX software.
- Line 241: Consider “In this study, the ecosystem was judged to be perturbed when the biomass indicator changed by 5% or more relative to the control simulation.”
RESULTS AND DISCUSSION
- Line 274: Describe what is meant by an “environmental receptor”.
- Figure 4: Please define the meaning of the different line colors. I can assume these are the river segments? Do the % biomass change values given under each sub-plot refer to the individual river segments? Please describe what (segment?) these numbers refer to (also do this for Fig. 5). In the methods text, the authors need to state that the relative change in biomass was evaluated 20 days after the simulated spill accident.
- Line 285: How was significance tested? Was there a statistical test? If so, please describe the statistical test. Or does significance refer back to the statement made at Methods Line 241 that “the criteria which judge the [ecosystem to be] perturbed or not was set as 5% of each indictor.” If there was no formal statistical test, please change wording to avoid using “significance” and replace with a qualitative term such as “strong”, “weak”, “substantial”, etc.
- Line 306: Remove the phrase “In conclusion” here. This makes the reader think this is the conclusion of the paper. Also, the authors could mention other potential sources of toluene spills, such as from pipelines, if they exist, or state that pipeline spills don’t exist. The authors could also state whether or not toluene spills are only ever episodic, short-term events or if there are potential long-term, chronic sources of pollution as from contaminated groundwater entering the river.
- Line 372: I think this is worded incorrectly and does not convey the meaning that the authors intend. The greater effect upon fish is not a TOXIC effect because they have a similar LC50 as the diatoms. The greater effect is due to ecological or trophic effects. Please consider “Although fishes and diatoms have similar LC50, the biomass decrease of fishes was more severe than that of the diatom. The ecological, or trophic, effects on the higher predator fish species are greater than the toxic effect of toluene upon producer diatoms.”
Line 374: Remove the word “Therefore”.
Line 377: Consider “…more sensitive than the toxic values would suggest.”
Line 379: If this is a correct statement, consider “This was because the physiology of each species in the food chain is different, and the impact of accidents is an opportunity to for faster-growing lower trophic level species to increase their biomass [19] when predation pressure is reduced.”
Line 382: “…similar level of sensitivity to toxicity…”
Line 384: “…but also to complementary effects of changes within the food web.”
Line 385: “…can provide more realistic estimates of biomass changes …”
Figure 7: If possible, please edit the symbol key to put a space between each of the five accident treatments.
Line 392: “From the point of view …”
Line 397: “…but fish were more sensitive to toluene spills …”
- Line 398: “Consider “Invertebrate and diatom species represented more than 90% of the total biomass in the ecosystem whereas their biomass change after the accident was relatively small …”
- Line 411: Just a question of interest. Did the authors run longer scenarios to see how long the ecosystem took to recover from the spill, if it ever did recover?
CONCLUSION
-Line 416: Delete “firstly”.
- Line 416 and throughout the paper: Consider changing “chemical accident” to “chemical spill”.
- Line 417: By “a provide”, do you mean “an established” or “a publically available”?
- Line 418: Needs rewording. Consider “In this study, we simulated the time of the spill as July 1. We assumed the water flow, biomass, and water quality (e.g. nutrient concentrations) to be stable, using the values observed on July 1, 2014. The actual water system is different because the state of the water body is constantly changing, but this is a short-term simulation over the duration of which we assume the physical water system to change little.”
SUPPLEMENTAL MATERIAL
- Tables S2, S3, and S4: Please state the source of the diatom data. Is it all from the Water Environment Information System? Please restate for each table the date of the observations, July 1, 2014.
- Table S5: Please define LC50 and EC50. LC50 means Lethal Concentration at 50% mortality? What does EC50 mean?
- Table S11: "Volueme" is misspelled.
- Figure S2: Are the two thumbtack images in the figure intentional?
